# Association between Periodontal Disease and Cognitive Impairment in Adults

**DOI:** 10.3390/ijerph20064707

**Published:** 2023-03-07

**Authors:** Najwane Said-Sadier, Batoul Sayegh, Raymond Farah, Linda Abou Abbas, Rania Dweik, Norina Tang, David M. Ojcius

**Affiliations:** 1College of Health Sciences, Abu Dhabi University, Abu Dhabi 59911, United Arab Emirates; 2Neuroscience Research Center (NRC), Lebanese University, Beirut 1533, Lebanon; 3INSPECT-LB (Institut National de Santé Publique, d’Épidémiologie Clinique et de Toxicologie-Liban), Beirut 1103, Lebanon; 4Department of Periodontics, University of the Pacific, San Francisco, CA 94103, USA; 5Department of Laboratory Medicine, Veterans Affairs Medical Center, San Francisco, CA 94121, USA; 6Arthur Dugoni School of Dentistry, University of the Pacific, San Francisco, CA 94103, USA

**Keywords:** cognitive impairment, dementia, oral pathogens, periodontitis, inflammatory biomarkers, antibodies, systematic review

## Abstract

Introduction: Periodontitis is a severe oral infection that can contribute to systemic inflammation. A large body of evidence suggests a role for systemic inflammation in the initiation of neurodegenerative disease. This systematic review synthesized data from observational studies to investigate the association between periodontitis and neuroinflammation in adults. Methods and materials: A systematic literature search of PubMed, Web of Science, and Cumulative Index to Nursing and Allied Health Literature (CINAHL) was performed for studies published from the date of inception up to September 2021. Search terms for the exposure “oral disease” and outcome “dementia”, “neuroinflammation” and “cognitive decline” were used. Study selection and data extraction were independently undertaken by two reviewers. The final eligible articles were included only if the exposure is periodontitis and the outcome is cognitive impairment or dementia or a topic related to this condition, and if the study was conducted in an adult population. The quality and risk of bias were assessed by Newcastle Ottawa Scale (NOS). Qualitative synthesis was used to narratively synthesize the results. Six cohort studies, three cross-sectional studies, and two case-control studies met the inclusion criteria. These eleven studies were only narratively synthesized. Meta-analysis was not performed due to the methodological heterogeneity of the studies. Results: The results of included studies show that chronic periodontitis patients with at least eight years of exposure are at higher risk of developing cognitive decline and dementia. Oral health measures such as gingival inflammation, attachment loss, probing depth, bleeding on probing, and alveolar bone loss are associated with cognitive impairment. The reduction of epidermal growth factor (EGF), interleukin 8 (IL-8), interferon γ-induced protein 10 (IP-10), and monocyte chemoattractant protein-1 (MCP-1) in addition to over expression of interleukin 1-β (IL-1β) are significant in patients suffering from cognitive decline with pre-existing severe periodontitis. Conclusions: All the included studies show evidence of an association between periodontitis and cognitive impairment or dementia and Alzheimer’s disease pathology. Nonetheless, the mechanisms responsible for the association between periodontitis and dementia are still unclear and warrant further investigation.

## 1. Introduction

Periodontal disease (PD) is a disease of periodontal tissues that results in attachment loss and destruction of alveolar bone [1]. The onset of disease is due to bacterial infection, mainly by *Porphyromonas gingivalis* (Pg) [2]. Until recently, it was thought that resident oral bacteria were only capable of generating disease confined within the oral cavity [3]. However, many studies have demonstrated that oral bacteria can contribute to systemic diseases, especially through their ability to cause systemic inflammation which is characterized by the induction of pro-inflammatory cytokines, chemokines, and exaggerated host immune responses [4,5]. One of the associations with systemic diseases is the one with cardiovascular diseases by promoting atheroma plaque development and progression [6]. More specifically, studies have raised the possibility that the oral microbiome could play a role in the onset of neuroinflammation in neurodegenerative diseases [7].

Neurodegenerative diseases such as Alzheimer’s disease (AD), Parkinson’s disease (PD), and amyotrophic lateral sclerosis (ALS) are characterized by progressive loss of neurons [8]. Three important hypotheses were set on the etiopathogenesis of AD, the “cholinergic hypothesis” by Bartus and colleagues [9], the “amyloid cascade hypothesis” described by Hardy and Allsop [10] and more recently the “inflammation hypothesis of AD” proposed by Krstic and Knuesel. In their hypothesis, Krstic and Knuesel stated that in late-onset AD—in contrast to the familial form of the disease—chronic inflammatory conditions may represent a major trigger of pathology by inducing phospho-tau-related cytoskeletal abnormalities and concomitant impairments of axonal transport [11]. This was not the first time where neuroinflammation involvement was discussed; more than a century ago, Alzheimer and his colleagues discussed the possibility that microorganisms may be involved in the formation of senile plaques, which are hallmarks of AD [12]. This study will focus on oral microorganisms that are associated with AD, such as *Porphyromonas gingivalis*. Research suggested that the association between periodontal pathogens and neuroinflammation could take place directly through blood and invasion of the blood–brain barrier, or indirectly by inducing an immune response. In support of the first hypothesis, a study was conducted using molecular and immunological techniques on the brains of AD patients. Six of seven periodontal *Treponema* species, namely *T. socranskii*, *T. pectinovorum*, *T. denticola*, *T. medium*, *T. amylovorum*, and *T. maltophilum*, were identified in the brains of AD patients using species-specific polymerase chain reaction [13]. On the other hand, another study confirmed that bacterial lipopolysaccharides (LPS) can activate Toll-like receptors (TLRs) expressed in glial cells, and thereby induce an inflammatory response due to the overexpression of pro-inflammatory cytokines such as IL-6, IL-1, TNF-α, and IFN-γ [7].

Future projections of the global population predict that by 2030, the world’s older population (≥65 years) will reach 1 billion, equivalent to 12% of the total global population; by 2050, it is estimated that the older population will represent 16.7% (or 1.6 billion) of the total global population [14]. This global shift in demographic characteristics toward an increase in the aging population presents not only a social and economic challenge, but also a great health concern as it is associated with the increasing prevalence of age-related diseases such as dementia [15].

In fact, the largest risk factor for developing Alzheimer’s disease is age. Recent projections indicate that by 2050, 1 in 85 individuals will be diagnosed with Alzheimer’s disease, which is the most common cause of dementia among the older population [16]. In addition, a workshop organized in 1999 demonstrated that the onset of chronic periodontitis (CP) is most commonly detected in older adults [17]. Moreover, epidemiological data demonstrated that periodontal disease affects over 10% and 50% of the young and older population, respectively [18]. Thus, it is important to understand better the correlation between periodontal inflammation and systemic complications. Such studies will provide an integrated picture revealing novel mechanisms underlying the etiopathogenesis of several human diseases such as dementia, bone loss, and other systemic diseases. In this review, we will summarize the evidence of association between periodontitis and cognitive impairment.

## 2. Materials and Methods

### 2.1. Design

This systematic review was performed in accordance with the Preferred Reporting Items for Systematic reviews and Meta-Analyses (PRISMA 2020) guidelines [19]. A statement of ethics was not required.

In this review, we aimed to summarize the evidence of association between periodontitis and cognitive impairment. We used the PICO framework (population, intervention (exposure), comparator, and outcome) in an effort to set the research question as follows: are adults aged 18 years or older (population) who suffer from periodontitis (exposure) at increased risk for cognitive impairment (outcome) compared with adults without periodontitis (comparator)?

### 2.2. Search Strategy

A systematic literature search of PubMed, Web of Science, and Cumulative Index to Nursing and Allied Health Literature (CINAHL) was performed for studies published from the date of inception up to September 2021. The databases were searched for eligible records using search terms pertaining to “oral pathogens”, “dementia”, and “neuroinflammation”. These search terms are ‘oral pathogens’, ‘oral bacteria’, ‘oral disease’, ‘oral health’, ‘mouth disease’, ‘gum disease’, ‘gum bacteria’, ‘periodontitis’, ‘gingivitis’, ‘*Porphyromonas gingivalis*’, ‘dementia’, ‘vascular dementia’, ‘peripheral neuroinflammation’, ‘central neuroinflammation’, ‘neurodegeneration’, ‘cognitive decline’, ‘memory loss’, ‘Alzheimer’s disease’, ‘Parkinson’, and ‘ALS’. Keywords were combined using the Boolean operators ‘OR’ and ‘AND’. The search was conducted with English language restrictions and was limited to human studies, while there was no limit for the year of publication. For more details of search formula refer to Appendix A.

### 2.3. Selection Criteria

All studies (including case-control, cohort, and cross-sectional studies) providing quantitative measures of the association between periodontal disease and dementia or neuroinflammation among adults were included in our review without regards to the region or country where the study was conducted. Eligible studies were those examining oral pathogenesis as exposure irrespective of the methods used to measure periodontitis or gingivitis. Also, no pre-specified restriction criteria for the assessment of dementia or cognitive impairment were adopted for study inclusion. Case reports, case series, randomized controlled trials, and studies related to the safety or effectiveness of treatments for oral pathogenesis, dementia, or neuroinflammation were not included. Moreover, studies on non-human subjects, subjects under 18 years of age, and patients already suffering from neurological disorders were excluded from the review.

### 2.4. Data Extraction and Synthesis

All references obtained were imported from online databases into Zotero. After automatically removing duplicates, identification of studies eligible for inclusion was performed by the two reviewers independently. Also, data extraction was done independently after full-text screening by the same reviewers to ensure inter-rater reliability and to prevent data entry errors. The characteristics of each study were summarized and tabulated to enable comparison and analysis across studies. Information was organized as basic study information (last author’s name, publication year), study design, basic sample characteristics (sample size, controls, and location), exposure, exposure measures, and results. Due to the variation in study elements, statistical combination of studies was not possible and only systematic synthesis was done. First, the design, setting, and study population were summarized. Then the results from included studies are simply described as well as possible and classified based on periodontitis assessment and diagnosis (clinical and laboratory parameters).

### 2.5. Methodological Quality Assessment

The methodological quality assessment of included studies was done by two independent reviewers using the Newcastle-Ottawa scale (NOS), which is a tool developed to assess the quality of nonrandomized studies [20]. A ‘star system’ has been followed in which a study is judged on three broad perspectives: the selection of the study groups, the comparability of the groups, and the ascertainment of either the exposure or outcome of interest for case-control or cohort studies, respectively. Such that, a study can be awarded a maximum of one star for each numbered item within the Selection and Outcome categories and a maximum of two stars can be given for Comparability. This scale was also modified for cross-sectional studies casting aside the items that do not fit with the design of such types of studies [21]. Consequently, according to the adapted NOS scale, a maximum of nine stars can be awarded for high-quality case-control and cohort studies while a maximum of six stars can be obtained for high-quality cross-sectional studies. Hence, case-control and cohort studies of six or more stars and cross-sectional studies of four or more stars are considered ‘high-quality’ studies [22]. Concerning the comparability of the groups studied, the main factors were the age and gender, while the second important factors were the comorbidities, knowing that the most common five comorbidities in Alzheimer’s disease were hypertension, osteoarthritis, depression, diabetes mellitus, and cerebrovascular disease [23]. Furthermore, for cohort studies, 10 years of follow-up was considered sufficient for the outcomes to occur. This decision was based on a study that demonstrated that patients with 10 years of CP exposure exhibited a higher risk of developing Alzheimer’s disease than unexposed groups [24]. Less than 10% of the loss to follow-up was considered acceptable.

## 3. Results

### 3.1. Study Selection

The electronic searches initially identified a total of 3693 records. After the exclusion of duplicates, 3189 studies were retained. Of these, 3151 abstracts did not meet the inclusion criteria, leaving 38 abstracts for further investigation. Full-text articles were obtained for all 38 studies and were read through for further evaluation. Of these, 11 studies met the full inclusion criteria and were included in this review. With regards to the classification of the studies according to study design, we obtained six cohort studies, three cross-sectional studies, and two case-control studies. Figure 1 illustrates the flowchart of the study selection process.

### 3.2. Study Characteristics

The 11 studies included in the review were classified as six cohort studies [24,25,26,27,28,29], three cross-sectional [30,31,32], and two case-control studies [33,34]. The majority of the studies (four of them) were conducted in the USA [26,27,28,32], two in Korea [25,34], two in Taiwan [24,29], and one each in Spain [33], Western Romania [30], and Denmark [31]. The number of participants enrolled in the studies ranged from 40 to 262,349 participants. Moreover, all the eligible studies included both men and women. All of the cohorts were followed over more than 10 years except for Stewart et al. with only five years of follow-up [28]. During the period of follow-up, some of the cohorts were diagnosed with periodontitis while others were not, and then the occurrence of cognitive impairment as in Sparks Stein et al. Stewart et al. [27,28], different types of dementia (Alzheimer-type, vascular, non-vascular, and over-all dementia) in Choi et al. [27], Demmer et al. [28], Tzeng et al. [29] or Alzheimer’s disease in Chen et al. [25], and Choi et al. [25] were evaluated to assess any possible association between periodontitis and the neurological outcome. A detailed summary of the study characteristics is available in Table 1.

Most of the studies used the Mini-Mental State Examination (MMSE) as a tool to systematically and thoroughly assess mental status. Other tests that were used are the immediate and delayed logical verbal memory test from the East Boston Memory Test, the three word registration/memory task (“apple”, “table” and “penny”), and the five serial subtractions by intervals of three [32]. A clinical psychologist performed Neuropsychological assessments for both delayed and immediate memory and distributive attention by using Rey Auditory Verbal Learning Test (RAVLT), Montreal Cognitive Assessment test (MOCA), Mini-Mental State Examination (MMSE), and Prague tests [30]. Two Wechsler Adult Intelligence Scale (WAIS) subtests, the Block Design Test (BDT), and Digit Symbol Test (DST) were used to examine working memory, which is found to be impaired in the case of pre-clinical and early Alzheimer’s disease [31]. Three studies used the diagnostics established by ICD-9-CM coding and ICD-10 codes for the diagnosis of Alzheimer’s disease and different types of dementia [24,25,29].

### 3.3. Risk of Bias in Studies

Using the Newcastle Ottawa Scale, the quality of the studies was assessed and tabulated. Appendix A shows the distribution of stars in each section (selection, comparability, and outcome) for all the studies. The cohort studies received six to nine stars out of nine, case-control studies received six and eight stars out of nine, while the cross-sectional studies were awarded between four and five stars out of a maximum of six stars. These results show that all the studies involved in this review are of high quality.

### 3.4. Association between Chronic Periodontitis and Cognitive Impairment

Since Periodontitis was defined in different studies either according to international classification of disease (ICD) or its clinical measurements by mouth examination, or by measurement of oral pathogens induced serological inflammatory biomarkers and antibodies, we decided to present the association of periodontitis and cognitive impairment based on the abovementioned criteria.

#### 3.4.1. Association between Chronic Periodontitis According to Its International Classifications and Cognitive Impairment

Clinical diagnosis of periodontal disease is made by the recognition of various signs and symptoms in the periodontal tissues which herald a departure from health [35]. Three cohort studies [24,25,29] included participants diagnosed with periodontitis based on the international classification of disease (ICD-9-CM: codes 523.4 and 523.1, ICD-10 code K05.3). In 2016, a 10-year clinical observational study conducted by Tzeng et al. revealed that patients with at least 8 years of chronic periodontitis and gingivitis exhibited a significantly higher risk of developing dementia than healthy groups. After adjusting for sex, age, monthly income, urbanization level, geographic region, and comorbidities, the HR for dementia was 2.54 (95% CI 1.297–3.352, *p* = 0.002) [29]. Furthermore, in 2017, Chen and colleagues aimed to identify the role of periodontitis in AD development. In their matched cohort study, they demonstrated that patients with a 10-year CP diagnosis exhibit an increased risk of developing AD (adjusted HR 1.707, 95% CI 1.152–2.528, *p* = 0.0077), regardless of co-morbidities, (Charlson comorbidity index) CCI score, or urbanization level. According to mediation analysis, the cerebrovascular disease was found to be a partial mediator between CP and AD, however, CP can also cause AD directly [24]. Moreover, in 2019, Choi et al. observed that, compared with non-chronic periodontitis participants, chronic periodontitis patients had elevated risk for overall dementia (adjusted HR = 1.06; 95% CI = 1.01–1.11) and Alzheimer’s disease (adjusted HR = 1.05; 95% CI = 1.00–1.11); a tendency toward increased vascular dementia risk among chronic periodontitis patients was also documented (adjusted HR = 1.10; 95% CI = 0.98–1.22). These results emphasize the findings from previous studies by showing that chronic periodontitis was significantly associated with dementia even after adjustments for lifestyle behaviors including smoking, alcohol consumption, and physical activity [25].

#### 3.4.2. Association between Chronic Periodontitis According to Its Clinical Measurement and Cognitive Impairment

The most recent classifications of periodontal diseases published by the American Academy of Periodontology (AAP) and case definitions were based on measurements of attachment level, probing depth, bone loss, and/or degree of inflammation [36]. In a large prospective cohort study done by Stewart et al., the associations between oral health measures and cognitive impairment and cognitive decline were investigated. Although most of the oral health measures were not associated with later cognitive impairment, gingival inflammation was the exception to be strongly associated with impairment and the only factor predicting cognitive decline. OR for this group compared with the remainder was 1.62, (95% CI = 1.09–2.42) [28]. In a population of black and white community-dwelling adults, the Periodontal Profile Class (PPC) was derived using attachment loss, probing depth, and bleeding on probing. Multivariable-adjusted hazard ratios (HRs) [95% CI] for incident dementia among participants with severe-PPC (vs. periodontal healthy) was 1.22 [1.01–1.47]. This result confirmed that periodontal disease, as defined by the periodontal profile class, is associated with a modestly increased risk for incident dementia and mild cognitive impairment [26]. Furthermore, in a cross-sectional study with age- and sex-matched case-control selection for elderly Koreans, Shin et al. assessed alveolar bone loss on dental panoramic radiographs to categorize the cumulative history of periodontitis (HOP) into three groups: normal, moderate periodontitis and severe periodontitis. His results demonstrated that participants with HOP were 2.1 times as likely to have cognitive impairment as those without (odds ratio (OR) = 2.14, 95% confidence interval (CI) = 1.04–4.41) and that the interaction effect of smoking and exercise on periodontitis highlighted the link. However, there was no significant dose–response effect on the association between cognitive impairment and the severity of periodontitis [34]. In another cross-sectional study, Kamer and colleagues examined the effect of periodontal inflammation on cognitive impairment by studying an older Danish population. The assessment of periodontal condition was conducted using clinical measures. Each tooth was assessed by inspection, probing, and scored using a Modified Community Periodontal Index (MCPI). Their results showed that the subjects with periodontal inflammation (PI) had significantly lower adjusted mean DST scores compared to subjects without PI (means 38.0 vs. 34.3; *p* = 0.02). However, for adjusted BDT, the significance held only for subjects with few missing teeth (31.9 vs. 30.8; *p* = 0.44) [31]. Consistent with these results, Hategan et al. showed for the first time that among young systemically healthy subjects, those with aggressive periodontitis (AGG_P) had impaired delayed episodic memory and learning rate compared to those with no signs of periodontitis (NL_P) or chronic mild/moderate periodontitis (CR_P). Comparing the means, delayed recall and immediate recall scores were lower in subjects with periodontitis (CR and AGG_P groups). In addition to memory, attention assessed by the Prague test was also compromised in both groups [30].

#### 3.4.3. Association between Periodontitis Induced Inflammatory Biomarkers and Cognitive Impairment

It is well known that pro-inflammatory cytokines play a pivotal role in cognitive decline, yet a clear cytokine profile for different stages of cognitive impairment and AD is yet to be presented due to inconsistent reports [37,38,39]. Montoya et al. demonstrated that serum concentrations of epidermal growth factor (EGF), interleukin 8 (IL-8), interferon γ-induced protein 10 (IP-10), and monocyte chemoattractant protein-1 (MCP-1) were significantly reduced in patients suffering from cognitive decline with pre-existing severe periodontitis [33]. This goes hand in hand with previous studies that showed downregulation of these proinflammatory cytokines and chemokines in cases with severe periodontitis [33,40]. This immune suppression coupled with cytokine and chemokine modulation aids *P. gingivalis*, a keystone in periodontal disease, to evade the host’s defenses by suppressing the innate immune response [41,42,43]. Furthermore, *P. gingivalis* exacerbates AD status, as it downregulates interferon-γ and impairs amyloid beta (Aβ) plaque clearance from the AD brain [44,45].

*P. gingivalis* infection promotes its own proliferation by abrogating EGF signaling pathways inducing oral bacterial dysbiosis favorable to *P. gingivalis* growth [46,47]. EGF is essential in memory recovery and reduction of cognitive impairment in multiple AD models. It has been shown to reduce APOE-4-induced cognitive impairment in an Aβ plaque load-independent manner [48,49]. Moreover, EGF enhances cerebrovascular function and rescues spatial learning and recognition memory as well as improves hippocampal neuronal function [50]. Abrogation of EGF signaling could be a possible explanation of how periodontal pathogens could potentially induce cognitive impairment. However, it is still unclear whether the effects of EGF signaling abrogation by periodontitis elicit cognitive impairment through these signaling molecules or through different EGF dependent pathways.

Another inflammatory biomarker to shed light on is interleukin 8 (IL-8). The expression of IL-8, a proinflammatory cytokine commonly degraded by *P. gingivalis* gingipain proteases [47], has been shown to be differentially altered in AD and dementia. Similar to the findings presented by Montoya et al., IL-8 is downregulated in dementia and cognitive impairment [51,52]. However, IL-8 is usually upregulated in AD, favoring neuroinflammation [53]. Due to some limitations indicated by Montoya et al., and the discrepancy between the mentioned studies, the recorded downregulation of IL-8 cannot be definitively considered as predictive of cognitive impairment, and these results must be interpreted with caution. IL-8 warrants further investigation as a candidate target molecule of *P. gingivalis*.

Previous studies reported elevated levels of the antimicrobial chemokine IP-10 (CXCL10) associated with neuroinflammation and cognitive impairment reported in Alzheimer’s disease and dementia [54,55,56,57]. Abrogation of IP-10 signaling leads to impaired T cell responses and proliferation, as well as decreased recruitment of CD4^+^ and CD8^+^ lymphocytes into the brain [58]. Interestingly, as part of the immunomodulation elicited by *P. gingivalis*, IP-10 has been demonstrated to be suppressed by the secretion of gingipain R without the need for direct cell-to-cell interaction [59,60]. Hence, further examination is needed to elucidate the effects of periodontitis on significant alteration of serum IP-10 levels in relation to cognitive impairment.

Furthermore, MCP-1, also known as CCL2, is also a proinflammatory chemokine that regulates host inflammatory responses and is usually upregulated by *P. gingivalis* lipopolysaccharide (LPS) [61,62,63]. However, Montoya et al. reported decreased MCP-1 serum levels in subjects suffering from periodontal disease with a high risk of cognitive impairment [33]. This finding has been corroborated by Takahachi et al., who reported that live *P. gingivalis* downregulated MCP-1 expression in cell culture [63]. Furthermore, MCP-1 levels have been shown to inconsistently affect Aβ status and deposition [64]. Although insignificant, elevated Aβ42/40 coupled with downregulation of MCP-1 had a higher magnitude of association with cognitive decline compared to elevated Aβ42/40 coupled with upregulation of MCP-1 [64]. Moreover, MCP-1 also plays a role in tau phosphorylation and the formation of neurofibrillary tangles [63,65] indicating a possible role of MCP-1 in cognitive decline and AD evolution. In light of what has been presented, the involvement of MCP-1 in cognitive decline resulting from severe periodontitis remains in question and warrants further investigation.

Lastly, several studies reported elevated interleukin 1-β (IL-1β) salivary levels, which further corroborate the role of periodontal disease in activating pro-inflammatory pathways [30,62,66]. This cytokine plays an important role in learning and memory and is known to be overexpressed in the AD brain, as it is secreted by activated microglial cells that are found in regions surrounding Aβ plaques [67]. This overexpression is to be both proinflammatory and neurotoxic and could be a key player in exacerbating disease status [68,69].

#### 3.4.4. Association between Serological Marker (IgG) of Periodontitis and Cognitive Impairment

Studies discussing the effect of antibodies specific to periodontal bacteria on cognitive impairment are rudimentary at best. Currently, the available data suggest that mild cognitive impairment is associated with pre-existing elevated serum IgG levels specific for *P. intermedia*, *F. nucleatum*, *P. gingivalis*, and *T. denticola*, with AD being strictly associated with serum anti-*P. gingivalis* and *T. denticola* IgG [27,32]. In this context, elevated serum IgG levels are associated with poor immediate verbal memory and impaired delayed recall. These findings remained robust after controlling for socioeconomic status, APOE ε4 status, and vascular disease [27,32]. With recent evidence suggesting that serum anti-*P. gingivalis* antibodies are associated with CSF total tau protein [70], and that periodontal pathogenic bacteria can be found in AD brains [71], it is still unclear whether IgGs specific for periodontal pathogens are cross-reactive with other targets that might exacerbate cognitive status. Thus, the aforementioned observations must be investigated further to verify if the released antibodies in response to periodontal pathogens are mediators of cognitive impairment, or whether the mere presence of these pathogens elicits cognitive impairment via other pathways.

## 4. Discussion

Periodontal diseases are infectious diseases of the oral cavity initiated mainly by gram-negative bacteria that trigger host immuno-inflammatory responses. One of the hallmarks of CP is the massive production of neutrophils, which has been reported to be associated with the acute host response to inflammation as well as the pathogenesis of chronic disease [72] Periodontal tissue destruction due to the release of degradative neutrophilic enzymes and/or cytotoxic factors, along with excessive IL-17 secretion by neutrophils recruited CD4^+^ T helper cells, contributes to potent inflammatory responses [73]. This local inflammatory process may induce a systemic inflammatory state. Based on the contribution of periodontal diseases in systemic inflammation and the potential role of systemic inflammation in neuroinflammation [74], understanding the association between oral diseases and dementia or neuroinflammation can uncover novel mechanisms underlying host–microbe homeostasis. A systematic review of the observational studies on this association was performed here, yielding six cohort, four cross-sectional, and two case-control studies. All the studies in our systematic review found scientific evidence of a link between exposure to periodontal pathogens and dementia or cognitive impairment [28,30,31,32,33,34,75] or more specifically neurodegenerative disease [24,25,27,76]. These findings are in agreement with previous studies that examined the contribution of periodontitis to AD onset and progression [77]. These studies found that the host response to subgingival periodontal pathogens engages both innate and adaptive immune responses, resulting in the alteration of local vasculature, generation of an inflammatory response, immune cell priming, and the secretion of pro-inflammatory cytokines including IL-1, IL-6, and TNF-α [78]. Another marker for inflammation is the high levels of C-reactive proteins (CRP): a study described a positive correlation between periodontitis and high levels of CRP, and patients with severe periodontitis have increased serum levels of CRP when compared with the unaffected control population [79].

Despite periodontal pathogens, chiefly *P. gingivalis*, being reported as causative agents of cognitive impairment and AD, the mechanism by which periodontal disease might influence the progression of cognitive impairment in AD patients is still unclear. One of the mechanisms that could lead to amplification of brain inflammatory molecules due to periodontitis may be via systemic circulation of the inflammatory mediators, either by crossing areas of the brain that lack a blood–brain barrier (BBB), or by entering the CNS in the areas with BBB by different pathways. The neural pathway is considered another possible route by which periodontal-derived cytokines could reach the brain. Interestingly, even if the elevated concentrations of these cytokines are only local and not systemic, they might still increase levels of brain cytokines [80]. Evidence from post-mortem analysis of AD brains shows that periodontal pathogens can locally infect AD brains via trigeminal nerve branches [71], and salivary biomarkers can mirror neurological diseases [81]. The amplification of brain cytokines could be done either by stimulating afferent fibers of peripheral nerves, leading to increased levels of brain cytokines, or can pass via channels or compartments associated with peripheral nerves.

Once in the brain, pro-inflammatory molecules might directly increase the local pro-inflammatory cytokine pool or indirectly by stimulating glial cells to produce pro-inflammatory cytokines including TNF–α, IL-1, IL-6, and inflammation reactive proteins such as CRP. Clinical studies showed that elevated CRP levels increased the risk of both developing AD [82] and of cognitive decline in various populations [83]. If glial cells are already primed or activated as in AD, stimulation would result in switching primed microglia to an aggressive pro-inflammatory phenotype contributing to aggravation neuroinflammation and neurodegeneration [84]. Moreover, *P. gingivalis* LPS can permeate neural cells and thus could potentially initiate signaling cascades that could exacerbate disease status or lead to AD progression from mild cognitive impairment [85]. LPS is also able to stimulate brain endothelial cells to release proinflammatory cytokines IL-6 and CCL2 [86]. Evidence from animal models further supports the role of LPS in memory and learning impairment, as LPS was able to induce Aβ pathology and neuroinflammatory responses [87].

In light of the findings discussed above, it is clear that periodontitis plays an important role in AD pathology and cognitive impairment. Yet the exact signaling cascade remains unclear and needs further investigation, especially since many contradictory results were obtained while assessing the findings presented by Montoya et al., who had already reported on these inconsistencies and unexpected results [33]. A major drawback is the paucity of data at present, which limits our ability to reach sound conclusions from the available studies. For example, as mentioned above, Montoya et al. only acquired one serum sample per patient, hence we are unable to objectively assess the effects of periodontitis on cognitive impairment. Montoya et al. also did not provide information concerning the causative agent of periodontitis in their sample, hence we cannot attribute the findings discussed above to one single pathogen. Thus, further longitudinal investigations are required with detailed follow-up to assess the exact effect of periodontitis on cognitive impairment. Another issue arose while assessing the data revealing that *P. gingivalis,* a prominent cause of periodontitis, induces oral bacteria dysbiosis and modulates the immune system to favor its survival. Thus, it remains unclear whether this immunomodulation elicited by *P. gingivalis* in the oral cavity, serum, or brain could influence the progression of cognitive impairment.

Our review has several other limitations which need to be taken into consideration when interpreting the results. Our electronic database search was not restricted concerning the year of publication. However, we assigned a search limiter for the English language, and this might have led to the exclusion of relevant eligible studies. Another limitation is due to the bidirectional relationship that has been proposed between AD and oral health. Data from a review assessing oral health in AD patients confirmed that the inability to self-care, including carrying out oral hygiene procedures, is one of the most probable causes of impaired oral health in AD [88]. Conversely, another study endorsed the view that chronic periodontitis could contribute to the clinical onset and progression of AD by several potential mechanisms [77]. This bidirectional effect confronts us with the problems of reverse causation and recalls subjectivity especially due to the involvement of cross-sectional and case-control studies in our review, where the temporal relationship between disease occurrence and outcomes cannot be fully established [89]. Moreover, the heterogeneity in terms of measures of diagnosis, outcome, and followed methodologies eliminated the chance of quantitative synthesis and necessitated a narrative synthesis of the results.

## 5. Conclusions

Our mouth is the gateway to our body. This oral–systemic connection means that oral pathogens could lead to systemic diseases such as heart attacks, stroke, and Alzheimer’s disease. The finding of our systematic review came out confirming this connection. It was clear that there is an association between periodontitis and dementia, and that patients with periodontitis are of greater risk of dementia and neurodegenerative diseases. However, the mechanisms responsible for this association should be studied more. Taken as a whole, oral health should not be separated from the rest body health; oral health must be prioritized in order to make serious steps toward better mental health.

## Figures and Tables

**Figure 1 ijerph-20-04707-f001:**
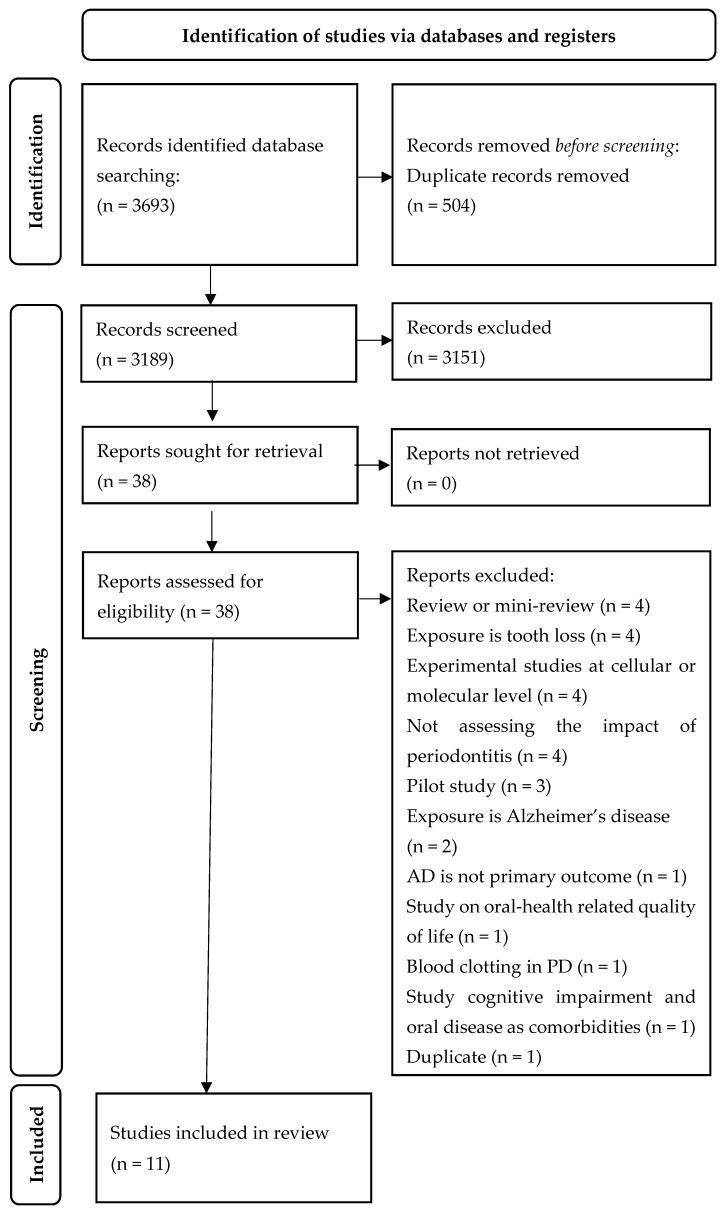
PRISMA flow diagram.

**Table 1 ijerph-20-04707-t001:** Characteristics of the reviewed studies and main findings.

Author	Study Design	Sample Characteristics (Size, Controls, Location)	Diagnosis	Exposure Biomarker(s)/Measure(s)	Outcome	Results
Montoya et al. [33]	Case-control	*n* = 309 Cases = 178 Controls = 131 Spain	periodontitis	Loss of attachment (degree of periodontitis) EGF, FGF-2, eotaxin, TGFα, G-CSF, GM-CSF, fractalkine, IFNγ, GRO, IL-10, MCP-3, MDC, IL-12 p70, IL-15, IL-17A, IL1-Ra, IL1-α, IL-1β, Il-5, IL-6, IL-7, IL-8, IP-10, MCP-1, MIP-1α, MIP-1β, TNF-α, TNF-β, and VEGF-A	mild cognitive impairment	EGF, fractalkine, IL-10, IL-5, IL-6, IL-7, IL-8, IP-10, MCP-1, MCP-1α, MCP-1β, and TNF-α were significantly and negatively associated with cognitive impairment, while G-CSF, IL1- Ra, and TNF-β were significantly and positively associated with this condition
Noble et al. [32]	Cross-sectional	*n* = 2355 USA	Periodontal disease caused by *P. gingivalis*	Serum *P gingivalis* IgG reported by ELISA	Cognitive impairment	The highest *P. gingivalis* IgG group was more likely to have poor delayed verbal memory and impaired subtraction
Shin et al. [34]	Case-control	*n* = 189 Cases = 65 controls = 124 Korea	history of periodontitis (HOP)	Radiographic alveolar bone loss (RABL)	Cognitive impairment	Participants with HOP were 2.1 times as likely to have cognitive impairment as those without
Sparks Stein et al. [27]	Cohort	*n* = 158 Cases = 81 Controls = 77 Kentucky, USA	Periodontal pathogens	IgG antibody levels to seven oral bacteria	Cognitive impairment	Antibody levels to *F. nucleatum* and *P. intermedia* were significantly increased at baseline serum draw in the patients with AD compared with control
Stewart et al. [28]	Prospective cohort study	N = 1171 Health ABC Study USA	Decline in oral health	number of teeth, number of occluding pairs of teeth, probing depth, loss of attachment, mean gingival index, mean plaque score, and number of sites with bleeding on probing	Cognitive impairment	Strong associations were found for gingival inflammation and 3MS decline measured from Year 1 to 3 or from Year 1 to 5
Tzeng et al. [29]	Retrospective matched cohort design.	N = 8828 Chronic periodontitis and gingivitis = 2207 Controls = 6621 Taiwan	Chronic periodontitis and gingivitis	ICD-9-CM codes: 523.4 (chronic periodontitis) and 523.1 (chronic gingivitis)	Dementia	Patients with newly diagnosed chronic periodontitis and gingivitis had an increased risk of dementia
C. K. Chen et al. [24]	Retrospective cohort study	N = 27,963 patients with CP = 9291 patients without CP = 18,672 Taiwan	Chronic periodontitis	Clinical assessment according to ICD-9-CM diagnostic criteria code 523.4	Alzheimer’s disease	10-year chronic periodontitis was associated with a 1.707-fold increase in the risk of developing Alzheimer’s disease
Choi et al. [25]	Retrospective cohort study	N = 262,349 (NHIS-HEALS) South Korea	Chronic periodontitis	International Classification of Diseases At least one treatment subgingival curettage, periodontal flap operation, gingivectomy, and odontectomy	Alzheimer’s disease	Chronic periodontitis patients had an elevated risk for overall dementia, Alzheimer’s disease, and vascular dementia
Demmer et al. [26]	Longitudinal cohort	n = 8275 (ARIC) United states	Periodontal disease	Probing depth, gingival recession and assessed bleeding on probing at six sites per tooth	Dementia and Mild Cognitive Impairment (MCI)	Periodontal disease was modestly associated with incident MCI and dementia
Hategan et al. [30]	Cross-sectional	N = 40 western Romania	Periodontitis	Radiographic assessment Saliva collection and cytokine assessment	cognitive dysfunction and Alzheimer’s disease	Delayed recall and immediate recall scores were lower in subjects with periodontitis Patients with aggressive periodontitis (AGG_P) had impaired delayed episodic memory and learning rate compared to mild/moderate periodontitis (CR_P) and no signs of periodontitis (NL_P)
Kamer et al. [31]	Cross-sectional	N = 152, Denmark	Periodontal inflammation	Each tooth was assessed by inspection, probing, and scored using a Modified Community Periodontal Index (MCPI)	Cognitive dysfunction	Subjects with periodontal inflammation had significantly lower DST and BDT (tests requiring memory) scores compared to those without periodontal inflammation

## Data Availability

Not applicable.

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
