# Peer review of "Association between Periodontal Disease and Cognitive Impairment in Adults"

_ijerph, 2023, doi:10.3390/ijerph20064707_

Round 1

Reviewer 1 Report (Previous Reviewer 1)

The manuscript has been revised well. I think this manuscript will be acceptable after some corrections have been done.

Author Response

Thank you for your kindly comments again.

Reviewer 2 Report (Previous Reviewer 3)

Thank you, now it is ok

Author Response

Thank you for your kindly comments again.

This manuscript is a resubmission of an earlier submission. The following is a list of the peer review reports and author responses from that submission.

Round 1

Reviewer 1 Report

In this study, Najwane Said-Sadier, et al evaluated the relationship among periodontitis and risk of dementia, such as Altzheimer’s disease, and so on, in review article. The authors reported that all the included study show evidence of an association between periodontitis dementia, however mechanisms are still unclear. This would be a useful study make contribution to people public health greatly. However, the manuscript and data could be still unmature and needed to be revised. Specific comments from this reviewer are as following.

In line 163, maybe you made a typographical error (Error! Reference source not found). And I recommend Figure 1 placed around “Study selection” part.

In line 166, you should describe “study characteristics” in more detail.

In line 180, you used reference no.35 as noun. I recommend [35] change ○○et al. it is same at line 187 (reference no. 29), line189 ([28],[29] , line 191 [26][27][30][25][26].

In line 207, maybe you made a typographical error (Error! Reference source not found).

In Table1, author is described in not only reference no. but also author’s name (○○, et al). It is same in Table 2.

Reviewer 2 Report

Dear authors,

I consider the review entitled “Association between Periodontal Disease and Risk of Dementia in Adults” I very interesting and with a profound impact in dental practice.

I consider that in the abstract, instead of ..... Articles without publication  date restriction were searched in PubMed, Web of Science, and CINAHL databases up to September 2021 ……  the authors must mention:

A systematic literature search of PubMed (including MEDLINE), Web of Science, and Cumulative Index to Nursing and Allied Health Literature (CINAHL) (with a filter used to exclude MEDLINE records) was performed for studies published from ? up to September 2021.

In Search strategy… The authors must specify….what was the time frame investigated? from when until September 2021?

Reviewer 3 Report

Dear Authors, this article about the association between periodontal disease and the risk of dementia in adults is very interesting. Knowing more about this "hidden" topic could really help clinicians and general population.

Some issues need to be solved before its final publication in the journal:

abstract: plebea divide abstract into introduction, materials and methods, results, conclusion.

Introduction: This part is well written; it would be nice to add a small chapter in the first lines, where you explain better how this inflammation pattern could cause many systemic disease, between the others also infertility. This papers could help: Ludovichetti FS, Signoriello AG, Artuso A, Zuccon A, Stellini E, Mazzoleni S. Periodontitis and female infertility: Is there a connection? Oral Dis. 2021 May;27(4):1069-1070. 

John V, Alqallaf H, De Bedout T. Periodontal Disease and Systemic Diseases: An Update for the Clinician. J Indiana Dent Assoc. 2016 Winter;95(1):16-23.

The materials and methods part and the results are well written and well performed.

The discussion part is also well written.

Please, add a "conclusion" chapter where you concisely explain the findings of your study
